biomaterials/developmental biology

barnacle, metamorphosis, biomineralization, calcite, growth rate

**Author for correspondence:**
Rebecca A. Metzler
e-mail: rmetzler@colgate.edu

# *Amphibalanus amphitrite* begins exoskeleton mineralization within 48 hours of metamorphosis

Rebecca A. Metzler[1], Jessica O'Malley[1], Jack Herrick[1], Brett Christensen[1], Beatriz Orihuela[2], Daniel Rittschof[3] and Gary H. Dickinson[3]

[1]Department of Physics and Astronomy, Colgate University, 13 Oak Drive, Hamilton, NY 13346, USA
[2]Marine Science and Conservation, Duke University Marine Laboratory, 135 Duke Marine Lab Road, Beaufort, NC 28516, USA
[3]Department of Biology, The College of New Jersey, 2000 Pennington Road, Ewing, NJ 08628, USA

RAM, 0000-0001-9196-9084

Barnacles are ancient arthropods that, as adults, are surrounded by a hard, mineralized, outer shell that the organism produces for protection. While extensive research has been conducted on the glue-like cement that barnacles use to adhere to surfaces, less is known about the barnacle exoskeleton, especially the process by which the barnacle exoskeleton is formed. Here, we present data exploring the changes that occur as the barnacle cyprid undergoes metamorphosis to become a sessile juvenile with a mineralized exoskeleton. Scanning electron microscope data show dramatic morphological changes in the barnacle exoskeleton following metamorphosis. Energy-dispersive X-ray spectroscopy indicates a small amount of calcium (8%) 1 h post-metamorphosis that steadily increases to 28% by 2 days following metamorphosis. Raman spectroscopy indicates calcite in the exoskeleton of a barnacle 2 days following metamorphosis and no detectable calcium carbonate in exoskeletons up to 3 h post-metamorphosis. Confocal microscopy indicates during this 2 day period, barnacle base plate area and height increases rapidly (0.001 mm$^2$ h$^{-1}$ and 0.30 µm h$^{-1}$, respectively). These results provide critical information into the early life stages of the barnacle, which will be important for developing an understanding of how ocean acidification might impact the calcification process of the barnacle exoskeleton.

# 1. Introduction

Barnacles, extraordinary arthropods dating back approximately 400 million years to the early Palaeozoic [1], have drawn the attention of prominent scientists, including Charles Darwin, owing to their diversity, tough exoskeleton, and the complex cement they secrete to adhere to substrates. Barnacles brood fertilized eggs and start life when the embryonated eggs hatch, releasing free-swimming nauplius larva. The nauplius form consists of six stages of development during which the organism feeds while swimming through the water column [2–4]. The cyprid stage of the barnacle occurs days to weeks after the nauplius stage, and the duration of this stage varies among species and with environmental conditions [2–4]. After undergoing a metamorphosis from the last naupliar stage to a cyprid, the cyprid uses its antennules to 'walk' around and assess surfaces. The cyprid does not feed and is dedicated to finding an appropriate substrate to attach to before undergoing metamorphosis into a sessile juvenile.

In adult acorn barnacles, the body tissues are mainly surrounded by a hard exoskeleton that consists of an operculum, that opens and closes to allow the organism within to feed, and parietal plates that surround the soft body. Many acorn barnacles also have a mineralized base plate that acts as a barrier between the organism and the substrate. Adult acorn barnacles grow by laterally extending the base plate while also vertically extending the parietal plates at the location where the parietal plates meet the base plate or substrate [5,6]. Living tissue is found within longitudinal and radial canals of the parietal and base plates, which play a role in shell plate growth [7,8]. The adhesive the adult secretes allows it to adhere to a single substrate for life.

In the majority of barnacles, the adult exoskeleton of the barnacle consists of the calcium carbonate polymorph calcite and a small percentage (less than 3 wt%) of organics [9,10]. The organic component largely consists of chitin matrix, acidic proteins and a sulphate-rich hydrogel [9–11]. In *Amphibalanus (=Balanus) amphitrite*, the species of barnacle studied here, the calcite component contains a small percentage of magnesium (1–2 mol%) and strontium (approx. 0.5 mol%), in addition to intracrystalline organics [11]. The calcite making up the *A. amphitrite* exoskeleton is more disordered than geological calcite on the atomic level [11]. In addition, the calcite crystals making up much of the exoskeleton vary in size, shape and orientation with little preferred orientation [11–13]. The combination of composition and structure results in a material that is harder and more resistant to fracture than geological calcite [14].

While the exoskeleton is vital to the sessile barnacle's survival, relatively little is known about the development and formation of this material, particularly during the initial hours after the animal metamorphoses from the cyprid larva. Prior assessments show the shell plates to be recognizable within the first hour after ecdysis and appear to stiffen over the next several hours [15]. By 4 day post-metamorphosis, shell increments within the plates are clearly visible [16]. Lacking from these earlier studies, however, is a systematic assessment of when and to what extent mineralization occurs. Thus, this work aims to identify the time at which mineralization of the exoskeleton occurs following metamorphosis, along with determining how the exoskeleton physically develops. We present a combination of confocal microscopy, scanning electron microscopy (SEM) and Raman spectroscopy data detailing these initial stages of barnacle exoskeleton formation. The data indicate that the exoskeleton starts as a soft organic matrix that begins calcifying less than 48 h after metamorphosis.

# 2. Material and methods

## 2.1. Experimental set-up

*Amphibalanus (=Balanus) amphitrite* cyprids were reared from field-collected adult barnacles at the Duke University Marine Laboratory following methods of [17,18]. A single larval batch was used for confocal and Raman experiments, while samples acquired from two separate batches were used for SEM experiments. Cyprids were settled in one of three growth environments: (i) glass confocal chamber slide; (ii) 24 well plate with a silicon wafer at the bottom of each well; or (iii) 24 well plate with a thin glass coverslip at the bottom of each well. The confocal chamber slides were used for confocal microscopy experiments while the silicon wafers and glass coverslips held within 24 well plate environments were used in Raman spectroscopy and SEM experiments. Both silicon wafers and glass slides were used as substrates to maximize settlement rates, in the case that cyprids showed a preference between these substrates. Each of the growth environments contained artificial seawater

(Instant Ocean, 32 psu). In half of the growth environments, a 140 mg l$^{-1}$ calcein (Sigma-Aldrich C0875; electronic supplementary material, figure S1) solution was added and prepared as in Jacinto et al. [19]. After placement in growth environments, cyprids were kept in a dark incubation chamber at 27.2°C. During daytime hours (8.00–17.00), the cyprids were monitored every hour to check for settlement (i.e. release of cyprid cement) and metamorphosis to the juvenile form. For barnacles in the 24 well plate environments, after metamorphosis was observed, juvenile barnacles were left to develop for specific time periods (1 h post-metamorphosis, 2 h post-metamorphosis, etc.). After this time period, juveniles were removed from the solution, washed with ethanol and acetone and placed in a −80°C freezer to preserve any amorphous calcium carbonate (ACC) that may be present for future analysis [20].

## 2.2. Confocal microscopy

A Zeiss LSM710 laser confocal microscope was used to examine three barnacles under a 20×, 0.8 NA objective following settlement in seawater containing calcein. Two of the barnacles were less than 16 h post-metamorphosis and one had committed to settling (i.e. it had secreted cement) but had not yet undergone a metamorphosis to the juvenile form. Images of the three barnacles were acquired every 10 min for the first 5.7 h and every 30 min for an additional 88.5 h. For each acquisition, a z-stack was automatically collected with spacing between each focal plane of 2.297 µm. The voxel dimensions were 1.38 µm × 1.38 µm × 2.3 µm and were used as a first order estimate of error. Calcein fluorescence was assessed using 488 nm light. As confocal imaging can provide highly detailed three-dimensional measurements [21], base area and height were measured from the image stacks taken on each of the three barnacles. Base area measurements were calculated by first determining the lowest in-focus image of the barnacle exoskeleton. The elliptical tool in IMAGEJ was then used to measure the area of the barnacle in the x- and y-plane at the constant focal plane (z) and successive time points. Error bars for area measurements were calculated based on 1.38 µm uncertainty in both the x- and y-dimensions. Height measurements were determined by calculating the number of z-steps in-between the highest in-focus exoskeleton image and the lowest in-focus exoskeleton image, and multiplying the said number by 2.297 µm, the spacing between each z-step; 2.3 µm error bars were used for the height measurement based on the z-scaling. Height measurements were made only when the entire parietal plate structure was within the depth of field.

## 2.3. Scanning electron microscopy

Barnacles were imaged with a JEOL JSM636OLV SEM after completion of Raman spectroscopy (see below) or after removal from the −80°C freezer. At room temperature, barnacle samples from 1 h post-metamorphosis to 6 day post-metamorphosis ((1) 1 h, (1) 3 h, (3) 1 day, (1) 2 day, (1) 3 day and (1) 6 day) were thawed, dried and coated with 10 nm of platinum and imaged in backscatter mode; this thawing procedure probably led to the collapse of the earlier stage exoskeletons. Energy-dispersive X-ray spectroscopy (EDS) was conducted with an Oxford X-max Silicon Drift X-ray Detector. EDS samples were embedded in epoxy resin while still attached to the substrate and orientated with their base plate down. Samples were polished with decreasing grit size silicon carbide papers until a cross section of the parietal plates were revealed. The final polish was done with 50 nm alumina slurry and the samples were coated with 10 nm of platinum prior to EDS [22]. On each sample (one individual per age point, with two individuals for the two day age point), 26–66 spectra were acquired in spot mode from several areas.

## 2.4. Raman spectroscopy

Five barnacles were selected for analysis with Raman spectroscopy. A 2 day old barnacle was used for initial assessments and was thus dry and at room temperature for several days prior to analysis. The remaining barnacles (1 h post-metamorphosis, 3 h post-metamorphosis, 3 day post-metamorphosis, 6 day post-metamorphosis) were kept frozen until the Raman spectroscopy experiments were conducted; experiments were done at room temperature and in air, thus by the time data were collected, samples were thawed and dried. Experiments were done at the Cornell Center for Materials Research (CCMR) on a Renishaw InVia Confocal Raman microscope with a 785 nm laser at 10% power and exposure times between 10 and 100 s. WIRE 4.1 was used to remove the baseline and peaks from cosmic rays, and average spectra (2–4 from different spots across a single organism). Spectra were compared to reference spectra for calcite [23], chitin [24] and ACC [25].

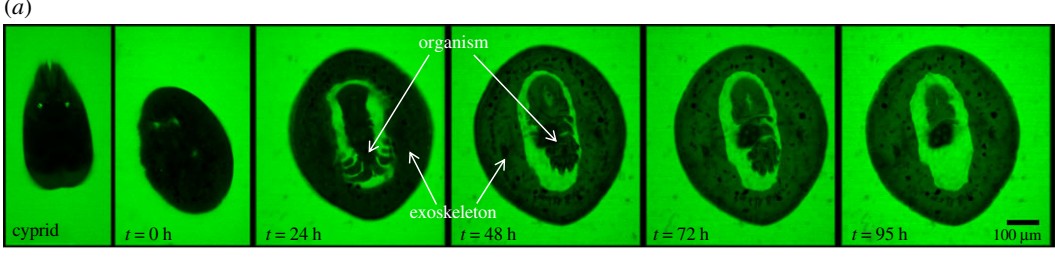

(a)

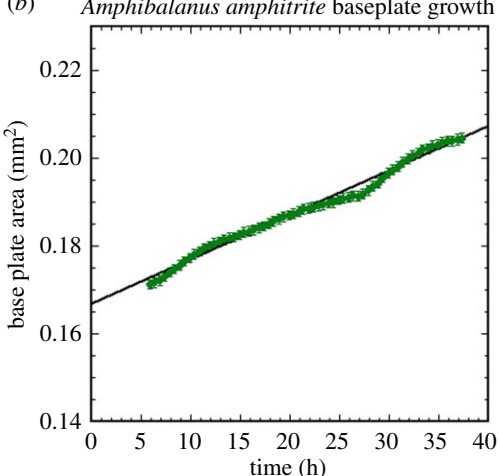

(b)

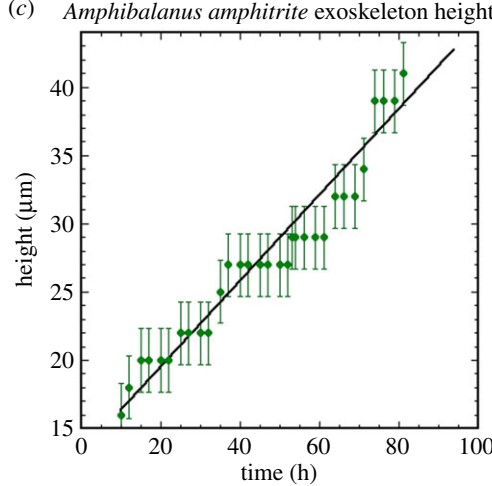

(c)

**Figure 1.** Confocal microscopy images were used to track a cyprid as it underwent metamorphosis and developed into a sessile juvenile in a calcein saltwater solution. (a) The first image, on the left, shows the organism before metamorphosis. Subsequent images show the barnacle body in the centre of the oval (dark shape), surrounded by saltwater (bright green) and the exoskeleton. The body is difficult to distinguish from the exoskeleton immediately following metamorphosis as the body and exoskeleton remain unstable and in motion for approximately 7 h post-metamorphosis. At 48 h, the exoskeleton appears green, indicating the incorporation of calcium (calcein) into the exoskeleton. (b) Measurements of the barnacle exoskeleton base area as a function of time (green dots), from 6 h post-metamorphosis onwards, indicate a steady, linear ($R^2 = 0.99$) increase over the first approximately 37 h post-metamorphosis. The slope of the line indicates a growth rate of 0.001 mm$^2$ h$^{-1}$, corresponding to a base diameter increase of 18 μm h$^{-1}$; error bars calculated based on 1.38 μm uncertainty in the $x$- and $y$-plane are shown. (c) Measurement of the exoskeleton height, from 6 h post-metamorphosis onwards, obtained by calculating the difference between the lowest in-focus exoskeleton field of view and the highest in-focus exoskeleton field of view, shows a linear increase in height with a rate of 0.30 μm h$^{-1}$; 2.3 μm error bars represent the uncertainty inherent in the $z$-plane measurement within the experimental set-up in the confocal microscope.

## 3. Results

Three barnacles grown in seawater with calcein were selected for tracking with laser confocal microscopy. Of the three barnacles, two were less than 16 h post-metamorphosis at the initiation of the confocal microscopy experiments and one had released its permanent adhesive but had not yet metamorphosed from a cyprid into a sessile juvenile. All three barnacles were tracked for a period of 94.2 h. Calcein binds to calcium and can be incorporated into biomineralizing structures [19,26]. When illuminated with 488 nm light, calcium bound calcein fluoresces green (555 nm).

Figure 1a shows confocal microscopy images of the growth and development of a barnacle from cyprid to sessile juvenile. The first image in the series is of the settled cyprid (in stage 2 of metamorphosis) [27]. The second image shows the barnacle immediately after metamorphosis, while the third image shows the barnacle 24 h after metamorphosis. In images from 24 h after metamorphosis to 95 h, the barnacle body is the dark region in the centre of the image and the exoskeleton is the dark ring surrounding the barnacle body. The surrounding calcein enriched saltwater solution can be seen fluorescing green in each of the images. Similarly, the bright green surrounding the barnacle body from 24 to 95 h is seawater enriched with calcein that is held within the mantle cavity. As can be seen in figure 1a, at the 48 h mark, the exoskeleton becomes more green in coloration, indicating the increased presence of calcein, or calcium (electronic supplementary material, figure S2).

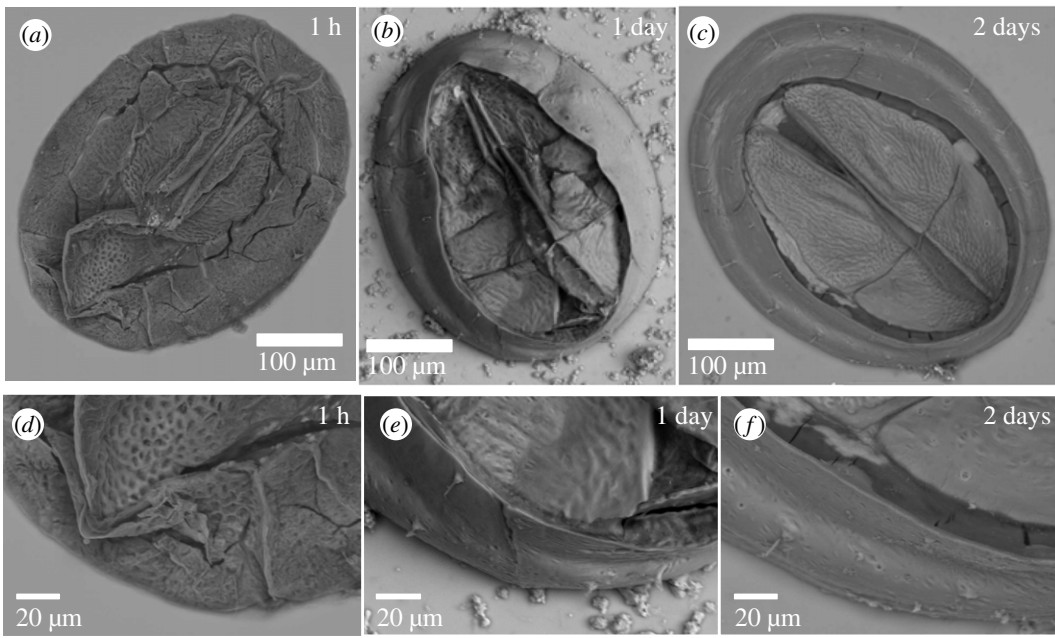

**Figure 2.** Scanning electron microscopy (SEM) images of 1 h, 1 day and 2 day old barnacles show rapid morphological changes. (*a*) The entire exoskeleton of a 1 h old barnacle collapses in on itself during the drying process. (*b*) The 1 day old barnacle exoskeleton exhibits some collapse with the operculum collapsing in on itself, while the parietal plates largely remain intact. (*c*) The 2 day old barnacle exoskeleton is unaltered by the drying process. (*d*) At higher magnification, the 1 h barnacle exoskeleton appears to have a rough surface that appears mesh-like in texture. (*e*) The 1 day old barnacle parietal plates appear smooth, while the operculum retains some of the mesh-like texture observed in the 1 h old barnacle. (*f*) The 2 day old barnacle operculum and parietal plates are both smooth with little mesh-like texture remaining.

In comparing the images in figure 1*a* a gradual change in the cross-sectional area of the barnacle can be observed over time. Figure 1*b* presents these base area measurements acquired from a single barnacle, showing a linear increase in the base area for the 37 h after metamorphosis. Prior to 6 h post-metamorphosis, the barnacle was still settling into its permanent position on the substrate, making base area measurements not possible. During the period of 6–37 h, the measured base area increased from $0.17\,mm^2$ to $0.20\,mm^2$ at a rate of $0.001\,mm^2\,h^{-1}$, corresponding to a diameter increase of roughly $18\,\mu m\,h^{-1}$. Similar base area growth rates were observed in the other two barnacles measured in the confocal microscope (electronic supplementary material, figure S3A). Measurements taken after the barnacles had spent 40 h under the confocal microscope (37 h post-metamorphosis for the barnacle in electronic supplementary material, figure S1B) were not used, as all three barnacles appeared to stop expanding their bases at that point in time, possibly owing to the heat from the confocal microscope (electronic supplementary material, figure S3B).

Similarly, by calculating the separation distance between the lowest in-focus exoskeleton confocal microscopy image and the highest in-focus exoskeleton confocal microscopy image at each time point, the height of the barnacle as a function of time was measured (figure 1*c*). The data of figure 1*c* show the height of the barnacle increases at a steady rate of $0.30\,\mu m\,h^{-1}$ after metamorphosis ($R^2 = 0.95$). A similar growth rate ($0.36$–$0.42\,\mu m\,h^{-1}$) was observed in the slightly older barnacles examined with confocal microscopy (electronic supplementary material, figure S4).

To supplement the confocal microscopy imaging, SEM was conducted on juvenile barnacles taken out of solution at specific time steps following metamorphosis. The overall morphology of the exoskeleton at 1 h, 1 day and 2 days after metamorphosis is shown in figure 2*a–c*. The 1 h exoskeleton is smaller than that of the 1 day or 2 day exoskeleton, while also appearing more fragile, as indicated by the collapsed appearance of the exoskeleton after drying at room temperature in air. The surface of the 1 h barnacle exoskeleton has a rough, mesh-like appearance that is highlighted in a higher magnification image (figure 2*d*). By 1 day post-metamorphosis, the exoskeleton is larger and more robust (figure 2*b*). While the operculum of the exoskeleton appears to have collapsed in on itself, the parietal plates remain intact. The surface of the 1 day barnacle exoskeleton is smoother than that of the 1 h exoskeleton, though the operculum still exhibits some of the mesh-like texture observed at 1 h

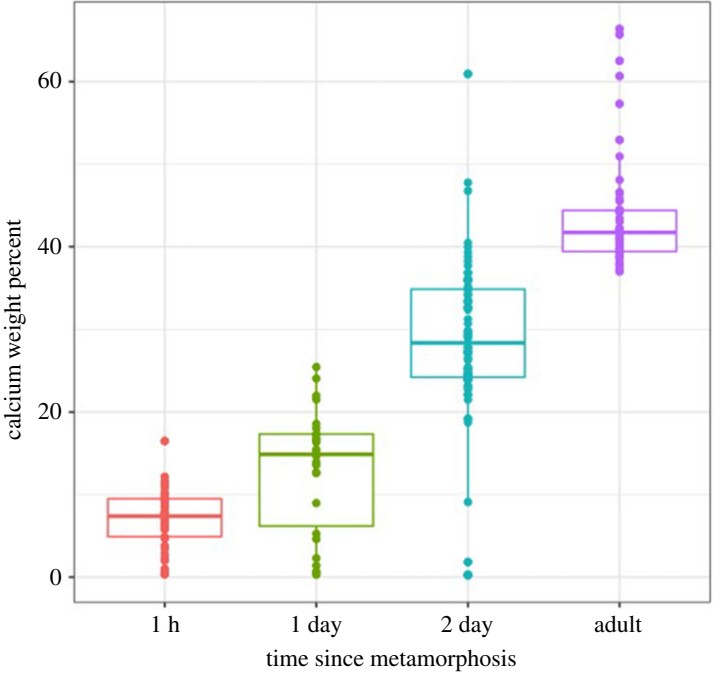

**Figure 3.** EDS of embedded and polished barnacle parietal plates show a steady increase in calcium content with age. The box-and-whisker plot indicates the median value for each age, with the dark line in the middle of the box, the upper and lower quartiles with the box boundaries, and the outermost measurements with the error bars or whiskers. The amount of calcium within each sample increases as a function of time with the median going from 8% at 1 h old to 16% at 1 day to 28% at 2 days. The adult barnacle had a median value of 44%. Data represent one individual sample per time point, with 26–66 measurements per sample. Graphs of the carbon and oxygen content as a function of time are in the supporting information (electronic supplementary material, figure S5).

(figure 2$e$). By 2 days post-metamorphosis (figure 2$c$), the exoskeleton looks like an adult exoskeleton, though smaller, with no evidence of exoskeleton failure upon drying. The surface of the 2 day exoskeleton is smooth with only a small amount of surface roughness (figure 2$f$).

EDS was conducted on the 1 h, 1 day and 2 day exoskeletons. EDS was done on polished parietal plate cross sections to remove topological effects. Figure 3 presents the weight percentage of calcium measured in the 1 h, 1 day and 2 day exoskeleton. The weight percentage of calcium measured in the exoskeleton of a several month-old barnacle was also obtained as an adult reference value. A total of 50–85 EDS measurements were taken on each of the samples with the median value represented as the darker line in the middle of each of the boxes (figure 3). The 1 h old barnacle had a median value of 7.8 wt% calcium, the 1 day 16.4 wt%, the 2 day 27.6 wt% and the adult 43.5 wt% (figure 3). As is shown in the plot (figure 3), in addition to having more calcium than the 1 h or 1 day barnacles, the 2 day old barnacle had the largest variability in the measured values for calcium weight percentage.

Raman spectroscopy of juvenile barnacles was used to confirm that mineralization of the exoskeleton occurs within the first 48 h, figure 4. Raman spectra were acquired from four barnacles (1 h, 3 h, 3 day, 6 day) that had been kept frozen prior to measurement. An additional Raman spectrum was collected on a single (2 day) barnacle that had been thawed and left at room temperature for several days prior to the Raman experiment. The 1 h, 3 h, 2 day and 3 day old barnacles were on silicon wafers, while the 6 day was on a glass coverslip. Calcite (in grey) and α-chitin (in black) Raman spectra are at the bottom of the graph (figure 4) as references for the mineral in adult *A. amphitrite* is calcite and the main organic component is chitin [9,10]. As can be seen in figure 4, the 1 h (green) and 3 h old (light blue) barnacles have none of the peaks associated with calcite or any other calcium carbonate polymorph. Rather, the main peaks seen in the 1 h and 3 h old barnacles are that of silicon (from the substrate, electronic supplementary material, figure S6), chitin and other organic molecules. The 2 day (slate blue) barnacle is the first to exhibit calcium carbonate peaks with calcite peaks observable at 282 cm$^{-1}$, 714 cm$^{-1}$ and 1086 cm$^{-1}$. The 3 day (dark blue) and 6 day (purple) barnacles also exhibit calcite peaks, with the 6 day barnacle exoskeleton exhibiting significantly fewer organic peaks than the 3 day barnacle. The 2 day barnacle spectrum also exhibits very few organic peaks.

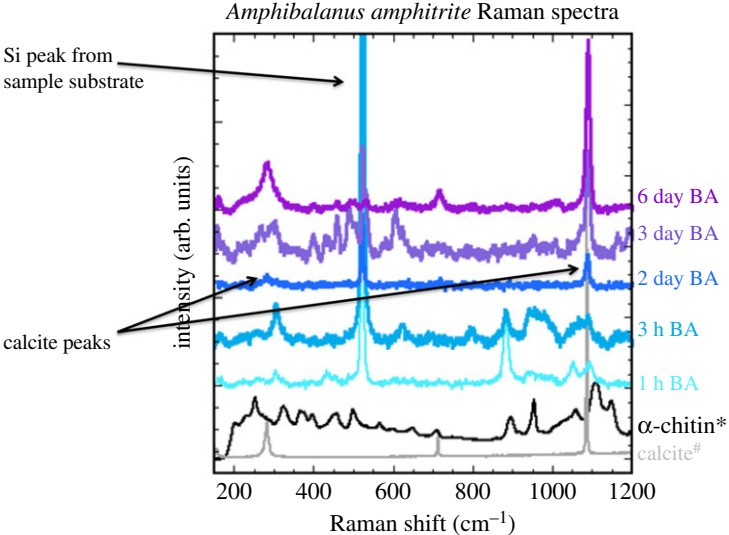

**Figure 4.** Raman spectroscopy of barnacles at different developmental stages indicates mineralization occurs when the barnacle is between 3 h and 2 days old. The 1 h (light teal) and 3 h old barnacle spectra (light blue) contain peaks consistent with chitin (black [24] and other organics, while calcitic (light grey [23] peaks appear in the 2 day (blue), 3 day (light purple) and 6 day old (dark purple) exoskeletons. No amorphous calcium carbonate peaks were observed [24,25].

## 4. Discussion

Metamorphosis from a mobile cyprid into a sessile juvenile barnacle is an intricate and multi-step process [6,27]. The organism not only must find a suitable substrate upon which to settle, the focus of the cyprid stage, but soon after settlement and metamorphosis, must also be able to withstand the environmental pressures that come with living in a harsh and variable environment (e.g. the intertidal region). The mineralized exoskeleton of the sessile barnacle plays a vital role in protecting the organism from predators and the changing hydration levels associated with the tide. Here, we find that the barnacle undergoes rapid base area growth during the measured time period (6–37 h) following metamorphosis, but does not begin mineralizing its exoskeleton until at least 3 h after metamorphosis.

The complex process through which the barnacle cyprid metamorphoses into a sessile juvenile ends with the animal exhibiting exoskeletal shell plates [27]. In watching the cyprid undergo metamorphosis and develop for more than 90 h, we were able to see the exoskeleton base grow steadily immediately following metamorphosis. In the first 37 h, the exoskeletal base area increases from $0.17 \, \text{mm}^2$ to $0.20 \, \text{mm}^2$, a rate of approximately $1000 \, \mu\text{m}^2 \, \text{h}^{-1}$ (figure 1$b$). The growth rates observed throughout the experimental period are higher than that observed in older juvenile barnacles, though consistent among the three barnacles measured here [5]. While increasing in diameter at the base, the exoskeleton also increases in height, with a steady $5–7 \, \text{nm} \, \text{h}^{-1}$ increase observed over the entire observation time (figure 1$c$). Thus, during the first 40 h following metamorphosis, the barnacle undergoes a period of remarkable growth—increasing not only in base width, but also in height.

Following 40 h under the confocal microscope, all three barnacles stopped expanding their bases. As the three barnacles were of different ages, we expect the plateau in growth was owing to experimental conditions (for example, heat or phototoxicity from the laser) rather than a natural physiological change in growth rate. Although the impact of fluorescence imaging on developing barnacle juveniles has not been assessed, reduced viability in cells and embryos in other species has been well documented [28].

Within the first few hours following metamorphosis, while this extraordinary growth is occurring, mineralization is notably absent. Despite mineralization of the exoskeleton being important for the protection of the organism, the onset of mineralization does not occur for several hours to days. While EDS data (figure 3) indicates calcium is present in the 1 h old barnacle, aligning with preliminary data of LeFourgey *et al.* [29], the Raman spectroscopy data present no peaks associated with calcium carbonate (figure 4). The calcium carbonate peaks do not appear in the Raman spectra until the barnacle is 2 days old (figure 4). Similar delays in mineralization have been observed in several developing mollusc shells with mineralization detected in the *Biomphalaria glabrata* snail shell 10–12 h

after the embryo began producing shell layers [30,31] and 8 h after initial shell layer detection in *Haliotis tuberculata* [32]. Likewise, in the crab *Callinectes sapidus*, calcium began to accumulate within the carapace 3 h after moulting, but calcite was not detected until 12 h post-moult [33]. In all cases, ACC is the first mineral detected [31,32].

The SEM images shown in figure 3 suggest there may be differences in the mineralization rate or onset of mineralization between the different exoskeletal plates. Observations are generally consistent with those shown by Glenner & Høeg [15]. The 1 h exoskeleton appears more fragile than the 1 day or 2 day exoskeletons as it has collapsed in on itself, probably when drying, in the SEM images. The 1 day parietal plates appear more robust than the 1 day operculum with the operculum collapsing in on itself and the parietal plates largely maintaining their shape. The entire 2 day exoskeleton maintains its shape. While the SEM images alone cannot verify whether mineralization has occurred, it seems likely there is a relationship between exoskeleton robustness and mineralization, and that the parietal plates may begin the mineralization process before the operculum. Future experiments assessing the time period between 3 h and 2 days will reveal whether there is a difference in mineralization between the operculum and parietal plates.

Thus far, the only calcium carbonate polymorph detected in the forming exoskeleton is calcite. While the adult exoskeleton of *A. amphitrite*, and most other sessile barnacles, consists entirely of calcite, most calcium carbonate biomineralizing organisms begin the mineralization process with ACC [34–37]. As we have not, as of yet, pinpointed the exact time at which biomineralization of the exoskeleton begins, it is still possible the exoskeleton mineralization starts with ACC. Furthermore, the way in which the calcite crystals are deposited within the juvenile exoskeleton remains unknown, leading to questions about at which point in time the juvenile exoskeleton fully resembles that of the adult barnacle in both structure and function.

# 5. Conclusion

The *A. amphitrite* exoskeleton forms at the end of the barnacle's metamorphosis from a cyprid to a sessile juvenile. The exoskeleton protects the organism within from tidal humidity variations and predators. During the first few hours following exoskeleton formation, or metamorphosis, the exoskeleton grows in both lateral and vertical dimensions while remaining completely organic with a mesh-like texture. After the first few hours, the exoskeleton becomes more robust, with a smoother texture. Within 2 days following metamorphosis, the exoskeleton looks much like an adult exoskeleton and has begun the process of mineralization with calcite detected within 3 days post-metamorphosis. The period between 3 h and 2 days needs to be explored in more detail to identify the exact onset of exoskeleton mineralization and the initial calcium carbonate polymorph deposited. The results of this study and future works will provide critical information into the life cycle of the barnacle, developing a baseline from which to explore the impact of ocean acidification on the biomineralization process of the barnacle exoskeleton.

Data accessibility. Raw data are uploaded to Dryad: https://doi.org/10.5061/dryad.brv15dv7g [38].

Authors' contributions. R.A.M., J.O., J.H. and B.C. carried out the growth and confocal experiments. R.A.M., J.O. and J.H. conducted the SEM, EDS and Raman experiments. J.O. and R.A.M. analysed the data. B.O. and D.R. provided samples. R.A.M. and G.H.D. conceived of the study, designed the study, coordinated the study and wrote the manuscript. All authors gave final approval for publication.

Competing interests. We declare we have no competing interests

Funding. This work was supported by the U.S. National Science Foundation under grant nos. DMR-1905619 to R.A.M. and DMR-1905466 to G.H.D. This work made use of Cornell Center for Materials Research Shared Facilities which are supported through the NSF MRSEC program (DMR – 1719875)

Acknowledgements. R.A.M. would like to thank Jason Meyers at Colgate University for his help with confocal microscopy experiments. R.A.M. is grateful for the support provided by Colgate University.

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
