## [Reviewer comments · Royal Society Open Science]

Review History

RSOS-200725.R0 (Original submission)

Review form: Reviewer 1

Is the manuscript scientifically sound in its present form?

Yes

Are the interpretations and conclusions justified by the results?

Yes

Is the language acceptable?

Yes

Do you have any ethical concerns with this paper?

No

Have you any concerns about statistical analyses in this paper?

No

Recommendation?

Accept with minor revision (please list in comments)

Comments to the Author(s)

This paper describes the exoskeleton mineralization of juvenile barnacles right after cyprid metamorphosis. The mineralization of the exoskeleton in these stages is less explored, making this study an important contribution to our knowledge of barnacle mineralization processes. SEM of 1h, 1d and 2d old barnacles showed a decreasing extent of exoskeleton collapse with time, suggesting a correlation between the extent of the collapse and the mineralization degree, which can affect exoskeleton stability. EDS measurements showed an increase in calcium content with time, with 28% calcium content after 2d. Confocal microscopy measurement of barnacles growing in calcein-labeled sea water at 0, 24h, 48h, 72h and 96h post metamorphosis show the appearance of exoskeleton labeling after 48h.

Specific comments:

- The base plate area and height were calculated based on the lowest in-focus image and the difference between the highest in-focus image and the lowest in-focus images of the exoskeleton. I have several comments regarding the height measurements: 1. What was the objective lens used (was not mentioned in the Methods)? Does the entire sample fit the depth of field? 2. What is the measurement error? Please add error bars. 3. Did you try to validate this method for measuring the height? Maybe by using an object with known dimensions and of a similar height range?
- It seems that the fixation/drying protocols used induced the collapse of the organic material, making its morphology and texture harder to observe under SEM. Optimizing the fixation protocol and using critical point drying can significantly improve the quality of the images and are recommended.
- Can you add to the manuscript/SI representative EDS data (summarized in Fig 3)? Was Mg detected, or does it only appear in later stages?
- What is the reason for the decrease in oxygen and carbon shown in Fig S5?
- Raman: It seems that most of the peaks that appear in the spectrum of the chitin reference are missing from the barnacle spectra, what is the reason for that? 895 cm⁻¹ is missing starting from 2d and 958 cm⁻¹ is missing from all barnacles. Since after 2d most of the barnacle is made of organic material (more than 70% based on the EDS measurements presented here), I would expect more pronounced peaks of chitin at these stages. Also, can you comment about the nature of the additional peaks that appear at the 3d sample (~600, 450-500 cm⁻¹)?
- For the Raman measurements, it will be beneficial to add the substrate spectrum, possibly in the SI
- Is there a reason why the Raman measurements were not conducted between the 3 h and 2-day development period? I am asking because if transient ACC indeed takes part in the exoskeleton mineralization, as suggested by the authors, it is possible that it is present during this time window. Especially due to the potentially ACC related peak in the 3d barnacle (1080 cm⁻¹, see Wang et al <https://onlinelibrary.wiley.com/doi/full/10.1002/jrs.3057>)
- The discussion starts with stating that “the barnacle undergoes rapid base area growth following metamorphosis, but does not begin mineralizing its exoskeleton until at least three hours after metamorphosis”. It is not clear to me whether the base growth rate for the first three hours after metamorphosis was measured or was extrapolated from later measurements (starting at 6h post metamorphosis)?

Review form: Reviewer 2

Is the manuscript scientifically sound in its present form?

Yes

Are the interpretations and conclusions justified by the results?

Yes

Is the language acceptable?

Yes

Do you have any ethical concerns with this paper?

No

Have you any concerns about statistical analyses in this paper?

Yes

Recommendation?

Accept with minor revision (please list in comments)

Comments to the Author(s)

Authors have explored mineralization process and rates of barnacle exoskeleton, especially during larval metamorphosis and very early stage of benthic life cycle. Results are certainly enhancing our understanding of this mineralization process in benthic invertebrates. The methodology and statistical analysis used are appropriate to address their objectives and there was no flaw. The results and the subject discussion are appropriate to publish in this journal. The manuscript is generally well written, so this reviewer recommends publication of this important manuscript after suitable minor revision.

Abstract: Implication of this study on climate change and biomaterials in the end of abstract, appears to be too ambitious and too broad. It would be easier for readers to link the results to project ocean acidification impacts on this critical life stage, e.g. rapid calcification and early barnacle skeleton is calcite based so is less vulnerable to OA shells (e.g. authors own previous papers and papers from Thiagarajan's group)

Introduction

Study objectives and the reason is not clearly and explicitly spelled out here - hope authors can pay attention and add couple of statements in the end of this section.

Materials methods

Page 4 - Line 23-26: How many culture jars used for this work, i.e. is cyprid from different culture jars mixed and used or only one culture jar? why shells were frozen at -80oC?

Reference: It would be good, if authors provide appropriate citation for the detailed methodology for each of those three techniques used in the study.

Results and discussion:

Well written. However, it would be great if authors link the observed barnacle mineralization process to ongoing and projects ocean acidification scenarios to gain new insights.

Decision letter (RSOS-200725.R0)

Dear Dr Metzler

On behalf of the Editors, we are pleased to inform you that your Manuscript RSOS-200725 "Amphibalanus amphitrite begins exoskeleton mineralization within 48-hours of metamorphosis" has been accepted for publication in Royal Society Open Science subject to minor revision in accordance with the referees' reports. Please find the referees' comments along with any feedback from the Editors below my signature.

Please submit your revised manuscript and required files (see below) no later than 7 days from today's (ie 11-Aug-2020) date. Note: the ScholarOne system will 'lock' if submission of the revision is attempted 7 or more days after the deadline. If you do not think you will be able to meet this deadline please contact the editorial office immediately.

on behalf of Dr Michael Doube (Associate Editor) and Kevin Padian (Subject Editor)
openscience@royalsociety.org

Associate Editor Comments to Author (Dr Michael Doube):

Dear Dr Metzler,

Thank you for your submission to RSOS. Two reviewers have seen your manuscript and are broadly supportive and positive about the work. They and I have a number of issues that must be addressed prior to final acceptance.

1. Raw data (confocal images, SEM images, EDS spectra, Raman spectra) must be uploaded to a public repository as a standard condition of publication in Royal Society journals <https://royalsociety.org/journals/authors/author-guidelines/#data>. This must include the full set of raw data (or of derived data where it is standard practice in the field), and not only selected examples for illustration.
2. The contextualisation should be more specific, including toning down the implications for climate change and biomaterials generally and rather focusing on ocean acidification which is narrower in scope and more directly relevant to biomineralisation. It would also help to spell out the study rationale and objectives near the end of the introduction.
3. Methodological details should be provided, including citations to methods used in the study, validation of measurement techniques including estimates of error, enumerating samples, detailed specifications of the instruments used and their state during measurement, potential interaction with specimen preparation artefacts such as shrinkage.
4. Please respond to the reviewers' specific queries including on the EDS and Raman spectra, barnacle base growth rate and decrease in oxygen and carbon in Fig S5.

Kind regards,
Michael Doube

Reviewer comments to Author:

Reviewer: 1

Comments to the Author(s)

This paper describes the exoskeleton mineralization of juvenile barnacles right after cyprid metamorphosis. The mineralization of the exoskeleton in these stages is less explored, making this study an important contribution to our knowledge of barnacle mineralization processes. SEM of 1h, 1d and 2d old barnacles showed a decreasing extent of exoskeleton collapse with time, suggesting a correlation between the extent of the collapse and the mineralization degree, which can affect exoskeleton stability. EDS measurements showed an increase in calcium content with time, with 28% calcium content after 2d. Confocal microscopy measurement of barnacles growing in calcein-labeled sea water at 0, 24h, 48h, 72h and 96h post metamorphosis show the appearance of exoskeleton labeling after 48h.

Specific comments:

- The base plate area and height were calculated based on the lowest in-focus image and the difference between the highest in-focus image and the lowest in-focus images of the exoskeleton. I have several comments regarding the height measurements: 1. What was the objective lens used (was not mentioned in the Methods)? Does the entire sample fit the depth of field? 2. What is the measurement error? Please add error bars. 3. Did you try to validate this method for measuring the height? Maybe by using an object with known dimensions and of a similar height range?
- It seems that the fixation/drying protocols used induced the collapse of the organic material, making its morphology and texture harder to observe under SEM. Optimizing the fixation protocol and using critical point drying can significantly improve the quality of the images and are recommended.
- Can you add to the manuscript/SI representative EDS data (summarized in Fig 3)? Was Mg detected, or does it only appear in later stages?
- What is the reason for the decrease in oxygen and carbon shown in Fig S5?
- Raman: It seems that most of the peaks that appear in the spectrum of the chitin reference are missing from the barnacle spectra, what is the reason for that? 895 cm⁻¹ is missing starting from 2d and 958 cm⁻¹ is missing from all barnacles. Since after 2d most of the barnacle is made of organic material (more than 70% based on the EDS measurements presented here), I would expect more pronounced peaks of chitin at these stages. Also, can you comment about the nature of the additional peaks that appear at the 3d sample (~600, 450-500 cm⁻¹)?
- For the Raman measurements, it will be beneficial to add the substrate spectrum, possibly in the SI
- Is there a reason why the Raman measurements were not conducted between the 3 h and 2-day development period? I am asking because if transient ACC indeed takes part in the exoskeleton mineralization, as suggested by the authors, it is possible that it is present during this time window. Especially due to the potentially ACC related peak in the 3d barnacle (1080 cm⁻¹, see Wang et al <https://onlinelibrary.wiley.com/doi/full/10.1002/jrs.3057>)
- The discussion starts with stating that “the barnacle undergoes rapid base area growth following metamorphosis, but does not begin mineralizing its exoskeleton until at least three hours after metamorphosis”. It is not clear to me whether the base growth rate for the first three hours after metamorphosis was measured or was extrapolated from later measurements (starting at 6h post metamorphosis)?

Reviewer: 2
 Comments to the Author(s)

Authors have explored mineralization process and rates of barnacle exoskeleton, especially during larval metamorphosis and very early stage of benthic life cycle. Results are certainly enhancing our understanding of this mineralization process in benthic invertebrates. The methodology and statistical analysis used are appropriate to address their objectives and there was no flaw. The results and the subject discussion are appropriate to publish in this journal. The manuscript is generally well written, so this reviewer recommends publication of this important manuscript after suitable minor revision.

Abstract: Implication of this study on climate change and biomaterials in the end of abstract, appears to be too ambitious and too broad. It would be easier for readers to link the results to project ocean acidification impacts on this critical life stage, e.g. rapid calcification and early barnacle skeleton is calcite based so is less vulnerable to OA shells (e.g. authors own previous papers and papers from Thiagarajan's group)

Introduction

Study objectives and the reason is not clearly and explicitly spelled out here - hope authors can pay attention and add couple of statements in the end of this section.

Materials methods

Page 4 - Line 23-26: How many culture jars used for this work, i.e. is cyprid from different culture jars mixed and used or only one culture jar? why shells were frozen at -80oC?

Reference: It would be good, if authors provide appropriate citation for the detailed methodology for each of those three techniques used in the study.

Results and discussion:

Well written. However, it would be great if authors link the observed barnacle mineralization process to ongoing and projects ocean acidification scenarios to gain new insights.

===PREPARING YOUR MANUSCRIPT===

- one version identifying all the changes that have been made (for instance, in coloured highlight, in bold text, or tracked changes);
- a 'clean' version of the new manuscript that incorporates the changes made, but does not highlight them. This version will be used for typesetting.

===PREPARING YOUR REVISION IN SCHOLARONE===

Author's Response to Decision Letter for (RSOS-200725.R0)

See Appendix A.

Decision letter (RSOS-200725.R1)

Dear Dr Metzler,

It is a pleasure to accept your manuscript entitled "Amphibalanus amphitrite begins exoskeleton mineralization within 48-hours of metamorphosis" in its current form for publication in Royal Society Open Science. The comments of the reviewer(s) who reviewed your manuscript are included at the foot of this letter.

on behalf of Dr Michael Doube (Associate Editor) and Kevin Padian (Subject Editor)
openscience@royalsociety.org

Associate Editor Comments to Author (Dr Michael Doube):
Associate Editor

Comments to the Author:

Thank you for making the revisions (and explaining where they have been impossible), which the reviewers and I requested.

There are a few final points that you and the RSOS editorial staff should address during the proofing stage:

1. the DOI to the data does not resolve. Make sure it does, and that all the image and spectral data are included, prior to final publication.
2. It would be more convenient for readers for you to provide the EDS table as a CSV file in your Dryad repository than a big table in the Supplementary information. Providing both is also OK - just upload the table to Dryad as a CSV or other non-proprietary format.
3. Confocal: there is no LSM70 by Zeiss. Did you mean LSM700, LSM710 or LSM780? Please list the NA of the objective as this is what sets the optical resolution. It can be found on the barrel of the objective next to the magnification as e.g. 20x/0.8 for an 0.8 NA objective. Finally - for information only - in confocal optical resolution in z is typically 5-10× larger than in xy (depending mostly on objective NA and pinhole diameter), while the precision of the stage z-stepper mechanism can be very fine indeed and much smaller than optical resolution. Next time take a look at the xz plane of a reconstructed confocal image stack.

Appendix A

Title: *Amphibalanus amphitrite* begins exoskeleton mineralization within 48-hours of metamorphosis

Rebecca A. Metzler^{1,*}, Jessica O'Malley¹, Jack Herrick¹, Brett Christensen¹, Beatriz Orihuela², Daniel Rittschof³, and Gary H. Dickinson³

1. Department of Physics and Astronomy Colgate University, 13 Oak Dr., Hamilton, NY 13346, USA

2. Duke University Marine Laboratory, Marine Science and Conservation, 135 Duke Marine Lab Rd., Beaufort, NC, 28516

3. Department of Biology, The College of New Jersey, 2000 Pennington Rd., Ewing, NJ 08628, USA

*Corresponding author: Rebecca A. Metzler, rmetzler@colgate.edu

Keywords: barnacle, metamorphosis, biomineralization, calcite, growth rate

Abstract

Barnacles are ancient arthropods that, as adults, are surrounded by a hard, mineralized, outer shell that the organism produces for protection. While extensive research has been conducted on the glue-like cement that barnacles use to adhere to surfaces, less is known about the barnacle exoskeleton, especially the process by which the barnacle exoskeleton is formed. Here we present data exploring the changes that occur as the barnacle cyprid undergoes metamorphosis to become a sessile juvenile with a mineralized exoskeleton. Scanning electron microscope (SEM) data show dramatic morphological changes in the barnacle exoskeleton following metamorphosis. Energy dispersive x-ray spectroscopy (EDS) indicates a small amount of calcium (8%) 1-hour post-metamorphosis that steadily increases to 28% by 2-days following metamorphosis. Raman spectroscopy indicates calcite in the exoskeleton of a barnacle 2-days following metamorphosis and no detectable calcium carbonate in exoskeletons up to 3-hours post-metamorphosis. Confocal microscopy indicates during this 2-day period, barnacle base plate area and height increases rapidly (0.001 mm²/hr and 0.30 μm/hr, respectively). These results provide critical information into the early life stages of the barnacle, which will be important for developing an understanding of how ocean acidification might impact the calcification process of the barnacle exoskeleton.

Introduction

Barnacles, extraordinary arthropods dating back ~400 million years to the early Paleozoic (1), have drawn the attention of prominent scientists, including Charles Darwin, due to their diversity, tough exoskeleton, and the complex cement they secrete to adhere to substrates. Barnacles brood fertilized eggs and start life when embryonated eggs hatch, releasing free-swimming nauplius larva. The nauplius form consists of six stages of development during which the organism

feeds while swimming through the water column (2-4). The cyprid stage of the barnacle occurs days to weeks after the nauplius stage, and the duration of this stage varies among species and with environmental conditions (2-4). After undergoing metamorphosis from the last naupliar stage to a cyprid, the cyprid uses its antennules to “walk” around and assess surfaces. The cyprid does not feed and is dedicated to finding an appropriate substrate to attach to before undergoing metamorphosis into a sessile juvenile.

In adult acorn barnacles, the body tissues are mainly surrounded by a hard exoskeleton that consists of an operculum, that opens and closes to allow the organism within to feed, and parietal plates that surround the soft body. Many acorn barnacles also have a mineralized base plate that acts as a barrier between the organism and the substrate. Adult acorn barnacles grow by laterally extending the base plate while also vertically extending the parietal plates at the location where the parietal plates meet the base plate or substrate (5, 6). Living tissue is found within longitudinal and radial canals of the parietal and base plates, which play a role in shell plate growth (7, 8). The adhesive the adult secretes allows it to adhere to a single substrate for life.

In the majority of barnacles, the adult exoskeleton of the barnacle consists of the calcium carbonate polymorph calcite and a small percentage (<3 wt%) of organics (9, 10). The organic component largely consists of chitin matrix, acidic proteins, and a sulfate-rich hydrogel (9-11). In *Amphibalanus (=Balanus) amphitrite*, the species of barnacle studied here, the calcite component contains a small percentage of magnesium (1-2 mol%) and strontium (~0.5 mol%), in addition to intracrystalline organics (11). The calcite making up the *A. amphitrite* exoskeleton is more disordered than geologic calcite on the atomic level (11). In addition, the calcite crystals making up much of the exoskeleton vary in size, shape, and orientation with little preferred orientation (11-13). The combination of composition and structure results in a material that is harder and more resistant to fracture than geologic calcite (14).

While the exoskeleton is vital to the sessile barnacle’s survival, relatively little is known about the development and formation of this material, particularly during the initial hours after the animal metamorphoses from the cyprid larva. Prior assessments show the shell plates to be recognizable within the first hour after ecdysis and appear to stiffen over the next several hours (15). By 4 days post metamorphosis, shell increments within the plates are clearly visible (16). Lacking from these earlier studies, however, is a systematic assessment of when and to what extent mineralization occurs. Thus, this work aims to identify the time at which mineralization of the exoskeleton occurs following metamorphosis, along with determining how the exoskeleton physically develops. We present a combination of confocal microscopy, scanning electron microscopy, and Raman spectroscopy data detailing these initial stages of barnacle exoskeleton formation. The data indicate that the exoskeleton starts as a soft organic matrix that begins calcifying less than 48 hours after metamorphosis.

Materials and Methods

Experimental set-up: *Amphibalanus (=Balanus) amphitrite* cyprids were reared from field-collected adult barnacles at the Duke University Marine Lab following methods of (17, 18). A single larval batch was used for confocal and Raman experiments, while samples acquired from two separate batches were used for SEM experiments. Cyprids were settled in one of three growth environments: 1) glass confocal chamber slide; 2) 24-well plate with a silicon wafer at the bottom of each well; or 3) 24-well plate with a thin glass coverslip at the bottom of each well. The confocal chamber slides were used for confocal microscopy experiments while the silicon wafers and glass coverslips held within 24-well plate environments were used in Raman spectroscopy and scanning electron microscopy experiments. Both silicon wafers and glass slides were used as substrates to maximize settlement rates, in the case that cyprids showed a preference between these substrates. Each of the growth environments contained artificial seawater (Instant Ocean, 32 psu). In half of the growth environments, a 140 mg/L calcein (Sigma-Aldrich C0875; SOM Figure S1) solution was added, prepared per Jacinto et al. 2015 (19). After placement in growth environments, cyprids were kept in a dark incubation chamber at 27.2°C. During daytime hours (8 am – 5 pm), the cyprids were monitored every hour to check for settlement (i.e. release of cyprid cement) and metamorphosis to the juvenile form. For barnacles in the 24-well plate environments, after metamorphosis was observed, juvenile barnacles were left to develop for specific time periods (1 hour post-metamorphosis, 2 hours post-metamorphosis, etc.). After this time period, juveniles were removed from solution, washed with ethanol and acetone and placed in a -80°C freezer to preserve any amorphous calcium carbonate that may be present for future analysis (20).

Confocal microscopy: A Zeiss LSM70 laser confocal microscope was used to examine three barnacles under a 20x objective following settlement in seawater containing calcein. Two of the barnacles were less than 16 hours post-metamorphosis and one had committed to settling (i.e. it had secreted cement) but had not yet undergone metamorphosis to the juvenile form. Images of the three barnacles were acquired every 10 minutes for the first 5.7 hours and every 30 minutes for an additional 88.5 hours. For each acquisition, a z-stack was automatically collected with spacing between each focal plane of 2.297 μm . The voxel dimensions were 1.38 μm x 1.38 μm x 2.3 μm and were used as a first order estimate of error. 488 nm light was used to obtain calcein fluorescence. As confocal imaging can provide highly detailed 3-D measurements (21), base area and height were measured from the image stacks taken on each of the three barnacles. Base area measurements were calculated by first determining the lowest in-focus image of the barnacle exoskeleton. The elliptical tool in ImageJ was then used to measure the area of the barnacle in the x- and y-plane at constant focal plane (z) and successive time points. Error bars for area measurements were calculated based on 1.38 μm uncertainty in both the x- and y- dimensions. Height measurements were determined by calculating the number of z-steps in-between the highest in-focus exoskeleton image and the lowest in-focus exoskeleton image, and multiplying said number by 2.297 μm , the spacing between each z-step; 2.3 μm error bars were used for the height measurement based on the

z-scaling. Height measurements were made only when the entire parietal plate structure was within the depth of field.

Scanning electron microscopy (SEM): Barnacles were imaged with a JEOL JSM6360LV SEM after completion of Raman spectroscopy (see below) or after removal from the -80°C freezer. At room temperature, barnacle samples from 1-hour post-metamorphosis to 6-days post-metamorphosis (1 1-hr, 1 3-hr, 3 1-day, 1 2-day, 1 3-day, and 1 6-day) were thawed, dried, and coated with 10 nm of platinum and imaged in backscatter mode; this thawing procedure likely led to collapse of the earlier stage exoskeletons. Energy dispersive x-ray spectroscopy (EDS) was conducted with an Oxford X-max Silicon Drift X-ray Detector. EDS samples were embedded in epoxy resin while still attached to the substrate and oriented with their base plate down. Samples were polished with decreasing grit size silicon carbide papers until a cross-section of the parietal plates were revealed. The final polish was done with 50-nm alumina slurry and the samples were coated with 10 nm of platinum prior to EDS (22). On each sample (one individual per age point), 26-66 spectra were acquired in spot mode from several areas.

Raman spectroscopy: Five barnacles were selected for analysis with Raman spectroscopy. A 2-day old barnacle was used for initial assessments and was thus dry and at room temperature for several days prior to analysis. The remaining barnacles (1-hour post-metamorphosis, 3-hour post-metamorphosis, 3-days post-metamorphosis, 6-days post-metamorphosis) were kept frozen until the Raman spectroscopy experiments were conducted; experiments were done at room temperature and in air, thus by the time data was collected, samples were thawed and dried. Experiments were done at the Cornell Center for Materials Research (CCMR) on a Renishaw InVia Confocal Raman microscope with a 785 nm laser at 10% power and exposure times between 10-100s. Wire 4.1 was used to remove the baseline and peaks from cosmic rays, and average spectra (2-4 from different spots across a single organism). Spectra were compared to reference spectra for calcite (23), chitin (24), and amorphous calcium carbonate (25).

Results

Three barnacles grown in seawater with calcein were selected for tracking with laser confocal microscopy. Of the three barnacles, two were less than 16 hours post-metamorphosis at the initiation of the confocal microscopy experiments and one had released its permanent adhesive but had not yet metamorphosed from a cyprid into a sessile juvenile. All three barnacles were tracked for a period of 94.2 hours. Calcein binds to calcium and can be incorporated into biomineralizing structures (19, 26). When illuminated with 488 nm light, calcium bound calcein fluoresces green (555 nm).

Figure 1A shows confocal microscopy images of the growth and development of a barnacle from cyprid to sessile juvenile. The first image in the series is of the settled cyprid (in Stage 2 of metamorphosis) (27). The second image shows the barnacle immediately after metamorphosis, while the third image shows the barnacle 24 hours after metamorphosis. In images from 24 hours after

metamorphosis to 95 hours, the barnacle body is the dark region in the center of the image and the exoskeleton is the dark ring surrounding the barnacle body. The surrounding calcein enriched saltwater solution can be seen fluorescing green in each of the images. Similarly, the bright green surrounding the barnacle body from 24-95 hours is seawater enriched with calcein that is held within the mantle cavity. As can be seen in **Figure 1A**, at the 48-hour mark, the exoskeleton becomes more green in coloration, indicating the increased presence of calcein, or calcium (**SOM Figure S2**).

Figure 1. Confocal microscopy images were used to track a cyprid as it underwent metamorphosis and developed into a sessile juvenile in a calcein saltwater solution. (A) The first image, on the left, shows the organism before metamorphosis. Subsequent images show the barnacle body in the center of the oval (dark shape), surrounded by saltwater (bright green) and the exoskeleton. The body is difficult to distinguish from the exoskeleton immediately following metamorphosis as the body and exoskeleton remain unstable and in motion for ~7 hours post-metamorphosis. At 48 hours, the exoskeleton appears green, indicating the incorporation of calcium (calcein) into the exoskeleton. (B) Measurements of the barnacle exoskeleton base area as a function of time (green dots), from 6 hours post-metamorphosis onwards, indicate a steady, linear ($R^2=0.99$) increase over the first ~37 hours post-metamorphosis. The slope of the line indicates a growth rate of $0.001 \text{ mm}^2/\text{hr}$, corresponding to a base diameter increase of $18 \text{ }\mu\text{m}/\text{hr}$; error bars calculated based on $1.38 \text{ }\mu\text{m}$ uncertainty in the x- and y- plane are shown. (C) Measurement of the exoskeleton height, from 6 hours post-metamorphosis onwards, obtained by calculating the difference between the

lowest in-focus exoskeleton field of view and the highest in-focus exoskeleton field of view, show a linear increase in height with a rate of 0.30 $\mu\text{m/hr}$; 2.3 μm error bars represent the uncertainty inherent in the z-plane measurement within the experimental set-up in the confocal microscope.

In comparing the images in **Figures 1A** a gradual change in the cross-sectional area of the barnacle can be observed over time. **Figure 1B** presents these base area measurements acquired from a single barnacle, showing a linear increase in base area for the 37 hours after metamorphosis. Prior to 6 hours post-metamorphosis, the barnacle was still settling into its permanent position on the substrate, making base area measurements not possible. During the period of 6-37 hours, the measured base area increased from 0.17 mm^2 to 0.20 mm^2 at a rate of 0.001 mm^2/hr , corresponding to a diameter increase of roughly 18 $\mu\text{m/hr}$. Similar base area growth rates were observed in the other two barnacles measured in the confocal microscope (**SOM Figure S3A**). Measurements taken after the barnacles had spent 40 hours under the confocal microscope (37 hours post-metamorphosis for the barnacle in **Figure S1B**) were not used, as all three barnacles appeared to stop expanding their bases at that point in time, possibly due to the heat from the confocal (**SOM Figure S3B**).

Similarly, by calculating the separation distance between the lowest in-focus exoskeleton confocal microscopy image and the highest in-focus exoskeleton confocal microscopy image at each time point, the height of the barnacle as a function of time was measured (**Figure 1C**). The data of **Figure 1C** shows the height of the barnacle increases at a steady rate of 0.30 $\mu\text{m/hr}$ after metamorphosis ($R^2 = 0.95$). A similar growth rate (0.36 – 0.42 $\mu\text{m/hr}$) was observed in the slightly older barnacles examined with confocal microscopy (**SOM Figure 4**).

To supplement the confocal microscopy imaging, scanning electron microscopy (SEM) was conducted on juvenile barnacles taken out of solution at specific time steps following metamorphosis. The overall morphology of the exoskeleton at 1-hour, 1-day, and 2-days after metamorphosis is shown in **Figure 2A-C**. The 1-hour exoskeleton is smaller than that of the 1-day or 2-day exoskeleton, while also appearing more fragile, as indicated by the collapsed appearance of the exoskeleton after drying at room temperature in air. The surface of the 1-hour barnacle exoskeleton has a rough, mesh-like appearance that is highlighted in a higher magnification image (**Figure 2D**). By 1-day post-metamorphosis, the exoskeleton is larger and more robust (**Figure 2B**). While the operculum of the exoskeleton appears to have collapsed in on itself, the parietal plates remain intact. The surface of the 1-day barnacle exoskeleton is smoother than that of the 1-hour exoskeleton, though the operculum still exhibits some of the mesh-like texture observed at 1-hour (**Figure 2E**). By 2-days post-metamorphosis (**Figure 2C**), the exoskeleton looks like an adult exoskeleton, though smaller, with no evidence of exoskeleton failure upon drying. The surface of the 2-day exoskeleton is smooth with only a small amount of surface roughness (**Figure 2F**).

Figure 2. Scanning electron microscopy (SEM) images of 1-hr, 1-day, and 2-day old barnacles show rapid morphological changes. (A) The entire exoskeleton of a 1-hr old barnacle collapses in on itself during the drying process. (B) The 1-day old barnacle exoskeleton exhibits some collapse with the operculum collapsing in on itself, while the parietal plates largely remaining intact. (C) The 2-day old barnacle exoskeleton is unaltered by the drying process. (D) At higher magnification, the 1-hr barnacle exoskeleton appears to have a rough surface that appears mesh-like in texture. (E) The 1-day old barnacle parietal plates appear smooth, while the operculum retains some of the mesh-like texture observed in the 1-hr old barnacle. (F) The 2-day old barnacle operculum and parietal plates are both smooth with little mesh-like texture remaining.

Energy dispersive x-ray spectroscopy (EDS) was conducted on the 1-hour, 1-day, and 2-day exoskeletons. EDS was done on polished parietal plate cross-sections to remove topological effects. **Figure 3** presents the weight percentage of calcium measured in the 1-hour, 1-day, and 2-day exoskeleton. The weight percentage of calcium measured in the exoskeleton of a several month-old barnacle was also obtained as an adult reference value. 50-85 EDS measurements were taken on each of the samples with the median value represented as the darker line in the middle of each of the boxes (**Figure 3**). The 1-hour old barnacle had a median value of 7.8 wt% calcium, the 1-day 16.4 wt%, the 2-day 27.6 wt%, and the adult 43.5 wt% (**Figure 3**). As is shown in the plot (**Figure 3**), in addition to having more calcium than the 1-hour or 1-day barnacles, the 2-day old barnacle had the largest variability in the measured values for calcium weight percentage.

Figure 3. Energy dispersive x-ray spectroscopy (EDS) of embedded and polished barnacle parietal plates show a steady increase in calcium content with age. The box-and-whisker plot indicates the median value for each age, with the dark line in the middle of the box, the upper and lower quartiles with the box boundaries, and the outermost measurements with the error bars or whiskers. The amount of calcium within each sample increases as a function of time with the median going from 8% at 1-hour old to 16% at 1-day to 28% at 2-days. The adult barnacle had a median value of 44%. Data represent 1 individual sample per time point, with 26-66 measurements per sample. Graphs of the carbon and oxygen content as a function of time are in the supporting information (SOM Figure S5).

Raman spectroscopy of juvenile barnacles was used to confirm that mineralization of the exoskeleton occurs within the first 48 hours, Figure 4. Raman spectra were acquired from four barnacles (1-hour, 3-hour, 3-day, 6-day) that had been kept frozen prior to measurement. An additional Raman spectrum was collected on a single (2-day) barnacle that had been thawed and left at room temperature for several days prior to the Raman experiment. The 1-hr, 3-hr, 2-day, and 3-day old barnacles were on silicon wafers, while the 6-day was on a glass coverslip. Calcite (in grey) and α -chitin (in black) Raman spectra are at the bottom of the graph (Figure 4) as references as the mineral in adult *A. amphitrite* is calcite and the main organic component is chitin (9, 10). As can be seen in Figure 4, the 1-hour (green) and 3-hour old (light blue) barnacles have none of the peaks associated with calcite or any other calcium carbonate polymorph. Rather, the main peaks seen in the 1-hour and 3-hour old barnacles are that of silicon (from the

substrate, Figure S6), chitin, and other organic molecules. The 2-day (slate blue) barnacle is the first to exhibit calcium carbonate peaks with calcite peaks observable at 282 cm^{-1} , 714 cm^{-1} , and 1086 cm^{-1} . The 3-day (dark blue) and 6-day (purple) barnacles also exhibit calcite peaks, with the 6-day barnacle exoskeleton exhibiting significantly fewer organic peaks than the 3-day barnacle. The 2-day barnacle spectrum also exhibits very few organic peaks.

Figure 4. Raman spectroscopy of barnacles at different developmental stages indicates mineralization occurs when the barnacle is between 3-hrs and 2-days old. The 1-hour (light teal) and 3-hour old barnacle spectra (light blue) contain peaks consistent with chitin (black (24) and other organics, while calcitic (light grey (23) peaks appear in the 2-day (blue), 3-day (light purple), and 6-day old (dark purple) exoskeletons. No amorphous calcium carbonate peaks were observed (24, 25).

Discussion

Metamorphosis from a mobile cyprid into a sessile juvenile barnacle is an intricate and multi-step process (6, 27). The organism not only must find a suitable substrate upon which to settle, the focus of the cyprid stage, but soon after settlement and metamorphosis, must also be able to withstand the environmental pressures that come with living in a harsh and variable environment (e.g. the intertidal region). The mineralized exoskeleton of the sessile barnacle plays a vital role in protecting the organism from predators and the changing hydration levels associated with the tide. Here, we find that the barnacle undergoes rapid base area growth during the measured time period (6-37 hours) following metamorphosis, but does not begin mineralizing its exoskeleton until at least three hours after metamorphosis.

The complex process through which the barnacle cyprid metamorphoses into a sessile juvenile ends with the animal exhibiting exoskeletal shell plates (27). In watching the cyprid undergo metamorphosis and develop for more than 90 hours, we were able to see the exoskeleton base grow steadily immediately following metamorphosis. In the first 37 hours, the exoskeletal base area increases from 0.17 mm² to 0.20 mm², a rate of ~1000 μm²/hr (Figure 1B). The growth rates observed throughout the experimental period are higher than that observed in older juvenile barnacles, though consistent among the three barnacles measured here (5). While increasing in diameter at the base, the exoskeleton also increases in height, with a steady 5-7 nm/hour increase observed over the entire observation time (Figure 1G). Thus, during the first forty hours following metamorphosis, the barnacle undergoes a period of remarkable growth – increasing not only in base width, but also in height.

Following 40 hours under the confocal microscope, all three barnacles stopped expanding their bases. As the three barnacles were of different ages, we expect the plateau in growth was due to experimental conditions (for example, heat or phototoxicity from the laser) rather than a natural physiological change in growth rate. Although the impact of fluorescence imaging on developing barnacle juveniles has not been assessed, reduced viability in cells and embryos in other species has been well documented (28).

Within the first few hours following metamorphosis, while this extraordinary growth is occurring, mineralization is notably absent. Despite mineralization of the exoskeleton being important for protection of the organism, the onset of mineralization does not occur for several hours to days. While energy dispersive x-ray spectroscopy (EDS) data (Figure 3) indicates calcium is present in the 1-hour old barnacle, aligning with preliminary data of LeFourgey et al. (29), the Raman spectroscopy data presents no peaks associated with calcium carbonate (Figure 4). The calcium carbonate peaks do not appear in the Raman spectra until the barnacle is 2-days old (Figure 4). Similar delays in mineralization have been observed in several developing mollusk shells with mineralization detected in the *Biomphalaria glabrata* snail shell 10-12 hours after the embryo began producing shell layers (30, 31) and 8 hours after initial shell layer detection in *Haliotis tuberculata* (32). Likewise, in the crab *Callinectes sapidus*, calcium began to accumulate within the carapace 3 hours after molting, but calcite was not detected until 12 hours post-molt (33). In all cases amorphous calcium carbonate (ACC) is the first mineral detected (31, 32).

The SEM images shown in Figure 3 suggest there may be differences in the mineralization rate or onset of mineralization between the different exoskeletal plates. Observations are generally consistent with those shown by Glenner and Høeg (15). The 1-hour exoskeleton appears more fragile than the 1-day or 2-day as it has collapsed in on itself, likely when drying, in the SEM images. The 1-day parietal plates appear more robust than the 1-day operculum with the operculum collapsing in on itself and the parietal plates largely maintaining their shape. The entire 2-day exoskeleton maintains its shape. While the SEM images alone cannot verify whether mineralization has occurred, it seems likely there is a relationship between exoskeleton robustness and mineralization, and that the parietal plates

may begin the mineralization process before the operculum. Future experiments assessing the time period between 3-hours and 2-days will reveal whether there is a difference in mineralization between the operculum and parietal plates.

Thus far, the only calcium carbonate polymorph detected in the forming exoskeleton is calcite. While the adult exoskeleton of *A. amphitrite*, and most other sessile barnacles, consists entirely of calcite, most calcium carbonate biomineralizing organisms begin the mineralization process with amorphous calcium carbonate (34-37). As we have not, as of yet, pinpointed the exact time at which biomineralization of the exoskeleton begins, it is still possible the exoskeleton mineralization starts with amorphous calcium carbonate. Furthermore, the way in which the calcite crystals are deposited within the juvenile exoskeleton remains unknown, leading to questions about at which point in time the juvenile exoskeleton fully resembles that of the adult barnacle in both structure and function.

Conclusions

The *A. amphitrite* exoskeleton forms at the end of the barnacle's metamorphosis from a cyprid to a sessile juvenile. The exoskeleton protects the organism within from tidal humidity variations and predators. During the first few hours following exoskeleton formation, or metamorphosis, the exoskeleton grows in both lateral and vertical dimensions while remaining completely organic with a mesh-like texture. After the first few hours, the exoskeleton becomes more robust, with a smoother texture. Within 2-days following metamorphosis, the exoskeleton looks much like an adult exoskeleton and has begun the process of mineralization with calcite detected within 3-days post-metamorphosis. The period between 3-hours and 2-days needs to be explored in more detail to identify the exact onset of exoskeleton mineralization and the initial calcium carbonate polymorph deposited. The results of this study and future works will provide critical information into the life cycle of the barnacle, developing a baseline from which to explore the impact of ocean acidification on the biomineralization process of the barnacle exoskeleton.

Acknowledgements

RM would like to thank Jason Meyers at Colgate University for his help with confocal microscopy experiments. RM is grateful for the support provided by Colgate University.

Funding

This work was supported by the U.S. National Science Foundation under Grant Nos. DMR-1905619 to RAM & DMR-1905466 to GHD. This work made use of Cornell Center for Materials Research Shared Facilities which are supported through the NSF MRSEC program (DMR – 1719875)

Author contributions

RAM, JO, JH, and BC carried out the growth and confocal experiments. RAM, JO, and JH conducted the SEM, EDS, and Raman experiments. JO and RAM analyzed the data. BO & DR provided samples. RAM and GHD conceived of the study, designed the

study, coordinated the study and wrote the manuscript. All authors gave final approval for publication.

References

1. Pérez-Losada M, Høeg JT, Crandall KA. Unraveling the Evolutionary Radiation of the Thoracican Barnacles Using Molecular and Morphological Evidence: A Comparison of Several Divergence Time Estimation Approaches. *Syst Biol.* 2004;53:244-64.
2. Yang Y. Larval Development of the Barnacle *Chinochthamalus scutelliformis* (Cirripedia: Chthamalidae) Reared in the Laboratory. *Journal of Crustacean Biology.* 2003;23:513-21.
3. Brown SK, Roughgarden J. Growth, Morphology, and Laboratory Culture of Larvae of *Balanus glandula* (Cirripedia: Thoracica). *Journal of Crustacean Biology.* 1985;5:574-90.
4. Tighe-Ford DJ, Power MJD, Vaile DC. Laboratory rearing of barnacle larvae for antifouling research. *Helgoländer wiss Meeresunters.* 1970;20:393-405.
5. Burden DK, Spillman CM, Everett RK, Barlow DE, Orihuela B, Deschamps JR, et al. Growth and development of the barnacle *Amphibalanus amphitrite*: time and spatially resolved structure and chemistry of the base plate. *Biofouling.* 2014;30:799-812.
6. Essock-Burns T, Gohad NV, Orihuela B, Mount AS, Spillmann CM, Wahl KJ, et al. Barnacle biology before, during and after settlement and metamorphosis: a study of the interface. *J of Experimental Biology.* 2017;220:194-207.
7. Gohad NV, Dickinson GH, Orihuela B, Rittschof D, Mount AS. Visualization of putative ion-transporting epithelia in *Amphibalanus amphitrite* using correlative microscopy: Potential function in osmoregulation and biomineralization. *J exp mar Biol Ecol.* 2009;380:88-98.
8. Checa AG, Salas C, Rodriguez-Navarro AB, Grenier C, Lagos NA. Articulation and growth of skeletal elements in balanid barnacles (Balanidae, Balanomorpha, Cirripedia). *Royal Society Open Science.* 2019;6:190458.
9. Barnes H, Klepal W, Mitchell BD. The Organic and Inorganic Composition of Some Cirripede Shells. *J exp mar Biol Ecol.* 1976;21:119-27.
10. Bourget E. Barnacle shells: Composition, structure, and growth. In: Southward AJ, editor. *Barnacle Biology.* Rotterdam: A. A. Balkema; 1987. p. 267-87.
11. Khalifa GM, Weiner S, Addadi L. Mineral and Matrix Components of the Operculum and Shell of the Barnacle *Balanus amphitrite*: Calcite Crystal Growth in a Hydrogel. *Cryst Growth Des.* 2011;11:5122-30.
12. Lewis AC, Burden DK, Wahl KJ, Everett RK. Electron Backscatter Diffraction (EBSD) Study of the Structure and Crystallography of the Barnacle *Balanus amphitrite*. *JOM.* 2014;66(1):143-8.
13. Nardone JA, Patel S, Siegel KR, Tedesco D, McNicholl CG, O'Malley J, et al. Assessing the Impacts of Ocean Acidification on Adhesion and Shell Formation in the Barnacle *Amphibalanus amphitrite* *Frontiers in Marine Science.* 2018;In press.
14. Raman S, Kumar R. Construction and nanomechanical properties of the exoskeleton of the barnacle, *Amphibalanus reticulatus*. *Journal of Structural Biology.* 2011;176:360-9.
15. Glenner H, Høeg JT. Scanning electron microscopy of metamorphosis in four species of barnacles (Cirripedia Thoracica Balanomorpha). *Mar Biol.* 1993;117:431-9.

16. Clare AS, Ward SC, Rittschof D, Wilbur KM. Growth Increments of the Barnacle *Balanus Amphitrite* Darwin (Cirripedia). . Journal of Crustacean Biology. 1994;14:27-35.
17. Rittschof D, Branscomb E, Costlow J. Settlement and behavior in relation to flow and surface in larval barnacles, *Balanus amphitrite* Darwin. J exp mar Biol Ecol. 1984;82:131-46.
18. Rittschof D, Clare AS, Gehart D, Mary SA, Bonaventura J. Barnacle *in vitro* assays for biologically active substances: toxicity and settlement inhibition assays using mass cultured *Balanus amphitrite amphitrite* Darwin. . Biofouling. 1992;6:115-22.
19. Jacinto D, Penteado N, Pereira D, Sousa A, Cruz T. Growth rate variation of the stalked barnacle *Pollicipes pollicipes* (Crustacea: Cirripedia) using calcein as a chemical marker. Sci Mar. 2015;79:117-23.
20. Politi Y, Metzler RA, Abrecht M, Gilbert B, Wilt FH, Sagi I, et al. Transformation mechanism of amorphous calcium carbonate into calcite in the sea urchin larval spicule. Proc Natl Acad Sci. 2008;105:17362-6.
21. Evans R, Harper IS, Sanson GD. Confocal imaging, visualization and 3-D surface measurement of small mammalian teeth. Journal of Microscopy. 2001;204:108-19.
22. Metzler RA, Rist R, Alberts E, Kenny P, Wilker JJ. Composition and Structure of Oyster Adhesive Reveals Heterogeneous Materials Properties in a Biological Composite. Adv Funct Mater. 2016;26:6814-21.
23. Lafuente B, Downs RT, Yang H, Stone N. The power of databases: the RRUFF project. . In: Armbruster T, Danisi RM, editors. Highlights in Mineralogical Crystallography. Berlin, Germany: W. De Gruyter; 2015. p. 1-30.
24. Engelson SB. SPECARRB [Available from: <http://www.models.life.ku.dk/~specarb/achitin.html>].
25. Tili MM, Amor MB, Gabrielli C, Joiret S, Maurin G, Rousseau P. Characterization of CaCO₃ hydrates by micro-Raman spectroscopy. Journal of Raman Spectroscopy. 2001;33:10-6.
26. Vidavsky N, Addadi S, Mahamid J, Shimoni E, Ben-Ezra D, Shpigel M, et al. Initial stages of calcium uptake and mineral deposition in sea urchin embryos. Proc Natl Acad Sci. 2014;111:39-44.
27. Maruzzo D, Aldred N, Clare AS, Høeg JT. Metamorphosis in the Cirripede Crustacean *Balanus amphitrite*. PLoS One. 2012;7.
28. Squirrell JM, Wokosin DL, White JG, Bavister BD. Long-term two-photon fluorescence imaging of mammalian embryos without compromising viability. . Nature Biotechnology. 1999;17:763-7.
29. LeFurgey A, Freudenrich CC, Wallace NR, Ingram P, Wilbur KM. The onset of biomineralization during cyprid to juvenile metamorphosis of the barnacle (*Balanus amphitrite amphitrite*). FASEB J. 1995;9:A639.
30. Bielfeld U, Becker W. Embryonic development of the shell in *Biomphalaria glabrata* (Say). Int J Dev Biol. 1991;35:121-31.
31. Marxen JC, Becker W, Finke D, Hasse B, Epple M. Early mineralization in *Biomphalaria glabrata*: Microscope and structural results. J Moll Stud. 2003;69:113-21.

32. Jardillier E, Rousseau M, Gendron-Badou A, Fröhlich F, Smith DC, Martin M, et al. A morphological and structural study of the larval shell from the abalone *Haliotis tuberculata*. *Mar Biol.* 2008;154:734-44.
33. Dillaman R, Hequembourg S, Gay M. Early pattern of calcification in the dorsal carapace of the blue crab, *Callinectes sapidus*. *Journal of Morphology.* 2005;263:356-74.
34. Weiss IM, Tuross N, Addadi L, Weiner S. Mollusc Larval Shell Formation: Amorphous Calcium Carbonate Is a Precursor Phase for Aragonite. *Journal of Experimental Zoology.* 2002;293:478-91.
35. Beniash E, Aizenberg J, Addadi L, Weiner S. Amorphous calcium carbonate transforms into calcite during sea urchin larval spicule growth. *Proc Biol Sci.* 1997;264:461-5.
36. Politi Y, Arad T, Klein E, Weiner S, Addadi L. Sea Urchin Spine Calcite Forms via a Transient Amorphous Calcium Carbonate Phase. *Science.* 2004;306:1161-4.
37. Auzoux-Bordenave S, Badou A, Gaume B, Berland S, Helléouet M-N, Milet C, et al. Ultrastructure, chemistry and mineralogy of the growing shell of the European abalone *Haliotis tuberculata*. *Journal of Structural Biology.* 2012;171:277-90.